# Trabectedin and Lurbinectedin Extend Survival of Mice Bearing C26 Colon Adenocarcinoma, without Affecting Tumor Growth or Cachexia

**DOI:** 10.3390/cancers12082312

**Published:** 2020-08-17

**Authors:** Giorgio Aquila, Andrea David Re Cecconi, Mara Forti, Roberta Frapolli, Ezia Bello, Deborah Novelli, Ilaria Russo, Simonetta Andrea Licandro, Lidia Staszewsky, Giulia Benedetta Martinelli, Laura Talamini, Laura Pasetto, Andrea Resovi, Raffaella Giavazzi, Eugenio Scanziani, Giorgia Careccia, Emilie Vénéreau, Serge Masson, Roberto Latini, Maurizio D’Incalci, Rosanna Piccirillo

**Affiliations:** 1Department of Neurosciences, Mario Negri Institute for Pharmacological Research IRCCS, 20156 Milan, Italy; giorgio.aquila@marionegri.it (G.A.); andrea.rececconi@marionegri.it (A.D.R.C.); forti9410@gmail.com (M.F.); giulibenny@hotmail.it (G.B.M.); 2Department of Oncology, Istituto di Ricerche Farmacologiche Mario Negri IRCCS, 20156 Milan, Italy; roberta.frapolli@marionegri.it (R.F.); ezia.bello@marionegri.it (E.B.); s.licandro@italfarmaco.com (S.A.L.); raffaella.giavazzi@marionegri.it (R.G.); maurizio.dincalci@marionegri.it (M.D.); 3Department of Cardiovascular Medicine, Istituto di Ricerche Farmacologiche Mario Negri IRCCS, 20156 Milan, Italy; deborah.novelli@marionegri.it (D.N.); russilar9@gmail.com (I.R.); lidia.staszewsky@marionegri.it (L.S.); serge.masson76@gmail.com (S.M.); roberto.latini@marionegri.it (R.L.); 4Department of Biochemistry and Molecular Pharmacology, Istituto di Ricerche Farmacologiche Mario Negri IRCCS, 20156 Milan, Italy; laura.talamini@marionegri.it (L.T.); laura.pasetto@marionegri.it (L.P.); 5Department of Oncology, Istituto di Ricerche Farmacologiche Mario Negri IRCCS, 24126 Bergamo, Italy; andrea.resovi@marionegri.it; 6Dipartimento di Medicina Veterinaria, Università di Milano, 20133 Milan, Italy; eugenio.scanziani@unimi.it; 7Mouse and Animal Pathology Lab (MAPLab), Fondazione UniMi, Università di Milano, 20139 Milan, Italy; 8Division of Genetics and Cell Biology, IRCCS San Raffaele Scientific Institute, 20132 Milan, Italy; careccia.giorgia@hsr.it (G.C.); venereau.emilie@hsr.it (E.V.)

**Keywords:** cancer cachexia, muscle atrophy, lurbinectedin, trabectedin, inflammation, splenomegaly

## Abstract

Trabectedin (ET743) and lurbinectedin (PM01183) limit the production of inflammatory cytokines that are elevated during cancer cachexia. Mice carrying C26 colon adenocarcinoma display cachexia (i.e., premature death and body wasting with muscle, fat and cardiac tissue depletion), high levels of inflammatory cytokines and subsequent splenomegaly. We tested whether such drugs protected these mice from cachexia. Ten-week-old mice were inoculated with C26 cells and three days later randomized to receive intravenously vehicle or 0.05 mg/kg ET743 or 0.07 mg/kg PM01183, three times a week for three weeks. ET743 or PM01183 extended the lifespan of C26-mice by 30% or 85%, respectively, without affecting tumor growth or food intake. Within 13 days from C26 implant, both drugs did not protect fat, muscle and heart from cachexia. Since PM01183 extended the animal survival more than ET743, we analyzed PM01183 further. In tibialis anterior of C26-mice, but not in atrophying myotubes, PM01183 restrained the NF-κB/PAX7/myogenin axis, possibly reducing the pro-inflammatory milieu, and failed to limit the C/EBP*β*/atrogin-1 axis. Inflammation-mediated splenomegaly of C26-mice was inhibited by PM01183 for as long as the treatment lasted, without reducing IL-6, M-CSF or IL-1β in plasma. ET743 and PM01183 extend the survival of C26-bearing mice unchanging tumor growth or cachexia but possibly restrain muscle-related inflammation and C26-induced splenomegaly.

## 1. Introduction

Cancer cachexia is a debilitating multifactorial syndrome, characterized by progressive deterioration and functional impairment of skeletal muscles, affecting about 80% of patients with advanced cancer [1]. Many of these patients also suffer from atrophy and dysfunction of diaphragm and cardiac muscles that may lead to respiratory [2] and heart failure [3], and thus to premature death.

Muscle wasting during cancer is mostly due to an imbalance between protein synthesis and degradation [4], often driven by increased systemic inflammation. Immune- or tumor cell-derived pro-inflammatory cytokines such as interleukin-6 (IL-6), tumor necrosis factor α (TNFα), high mobility group 1 (HMGB-1) [5] or IL-1β [6], trigger muscle wasting by inducing the nuclear factor kappa-light-chain-enhancer of activated B cells (NF-κB), Janus kinase/signal transducers and activators of transcription 3 (JAK/STAT3) or Forkhead box-containing subfamily O3 (FoxO3) pathways [7,8,9,10]. In atrophying muscles, NF-κB overactivation in turn leads also to abnormal accumulation of PAX7, a transcription factor driving muscle regeneration in normal circumstances but failing to do so during cancer cachexia [11]. CCAAT/enhancer binding protein *β* (C/EBP*β*) is a transcription factor shown to inhibit myogenesis during cancer cachexia [12], and it is induced in myoblasts by IL-1β [13]. Instead, FoxO3 and STAT3 are transcription factors that in muscles promote the expression of atrophy-related genes, namely atrogenes, such as atrogin-1 and MuRF1, two muscle-specific ubiquitin ligases that promote protein degradation via the proteasome [14,15,16]. Inhibition of C/EBP*β* has been shown to be useful to obviate muscle wasting during cancer cachexia through inhibition of solely atrogin-1 [17]. Despite the broad knowledge of the inflammation-mediated mechanisms underlying this pathology, no effective therapy against cancer cachexia is yet available.

Lurbinectedin (PM01183) is a synthetic alkaloid derivative of trabectedin (ET743), a marine drug approved for the treatment of advanced soft tissue sarcoma and ovarian cancer [18]. PM01183 is currently under investigation in a multicenter phase II clinical trial to assess its activity against several advanced solid tumors, such as neuroendocrine tumors, endometrial carcinoma and BRCA 1/2-associated metastatic breast carcinoma (NCT02454972) and under phase II evaluation in patients with small cell lung cancer [19]. Besides being an alkylating antineoplastic agent, PM01183 shares with trabectedin the ability to affect not only cancer cells but also the tumor microenvironment (reviewed in [20]). We recently reported that PM01183, at higher doses, affects the inflammatory tumor milieu via a direct cytotoxic effect on mononuclear phagocytes, whereas at lower doses it affects the adhesion and migration of monocytes, limiting their production of pro-inflammatory cytokines, such as IL-6, IL-8 and monocyte chemoattractant protein-1 (MCP-1) [21]. Although we reported in vitro and in vivo that both PM01183 and ET743 display comparable efficacy in affecting the tumor microenvironment [21,22], previous phase I-II clinical studies showed PM01183 as less toxic and better tolerated than its natural lead compound [19,23], suggesting pharmacological advantages over trabectedin for patients with advanced cancer.

High levels of pro-inflammatory cytokines are seen in various cancers causing rapid body wasting with muscle and cardiac tissue depletion, with or without loss of fat mass (i.e., cachexia) [24]. Although both of these anti-cancer drugs counteract the production of detrimental cytokines, including IL-6, IL-8 and MCP-1 [21,25], neither ET743 nor PM01183 has yet been evaluated as able to limit the progression of cancer cachexia. We previously described a significant association between weight gain and increased survival in patients with recurrent advanced soft tissue sarcoma following trabectedin treatment [26], indicative of an anti-cachexia activity of this drug. These effects might result from the downregulation of circulating inflammatory cytokines and/or from a direct effect of such drugs on skeletal muscle or cancer cells, as already shown both in vitro and in vivo for trabectedin [27].

In the present study, we explored the anti-cachexia potential of ET743 and PM01183 in mice bearing the C26 adenocarcinoma, which displayed high circulating levels of inflammatory cytokines, such as IL-6 [28], acute phase response activation [29] and splenomegaly [30]. We found that both drugs greatly extended the survival of C26-bearing mice, without altering tumor growth rate (contrarily to what other drugs or diet do [31,32]), body weights over time or daily food intake. We focused our further analyses at multi-organ level on PM01183 because its effects on survival of cachectic animals were greater than ET743. Thus, we analyzed multiple tissues, such as fat, muscle, heart and spleen of these mice also through sophisticated technologies (micro-computerized tomography and ultrasound-based imaging). Overall, we found that both the NF-κB/PAX7-related muscle inflammation and C26-induced splenomegaly were restrained by PM01183 for as long as the treatment lasted, with no change in the plasma levels of the main inflammatory cytokines, such as IL-6, IL-1β and macrophage colony-stimulating factor (M-CSF).

## 2. Results

### 2.1. Trabectedin and Lurbinectedin Impressively Increase the Survival of C26 Tumor-Bearing Mice

We investigated the anti-tumor and the anti-cachexia potential of ET743 and its analog PM01183 in mice inoculated with C26 colon adenocarcinoma. The administration of either drug failed to limit the progression of this tumor (Figure 1A), the optimal treated/control (T/C) ratio being 63% (day 19) and 71% (day 19) for ET743 and PM01183, respectively, with no statistically significant differences between the rates of tumor growth. C26-bearing mice (C26-mice) showed body weight (BW) loss by day 10–13 from tumor implant, compared to PBS-injected mice (PBS-mice) (Figure 1B). They were euthanized about 30 days from tumor injection, when animals displayed at least four out of five signs of distress (loss of mobility, kyphosis, ruffled fur, dehydration, tremor) or when more than 20% of BW was lost in 72 h or when the tumor showed initial signs of ulceration (Figure 1C). Notably, PM01183 treatment (from day 3 to day 21, 0.07 mg/kg, three times a week for three weeks) markedly extended the lifespan of C26-mice (Figure 1C) from a median survival time of 20 days (range day 10–31) to a median survival time of 37 days (range day 14–41). Similar data were obtained in C26-mice treated with ET743 (from day 3 to day 21, 0.05 mg/kg, three times a week for three weeks), which also displayed an increased median survival time of 26 days (range day 14–49) (Figure 1C). In both cases, there was a striking delay in the onset of signs of distress of C26-mice without affecting their BW and/or food intake (Figure 1B,D).

We found that C26 hosts displayed all lung metastases at sacrifice even if they differed from each other for the number (ranging from 1 to 51, *n* = 10 mice). Similarly, the PM-treated C26 hosts displayed all lung metastases at sacrifice (ranging from 1 to 81, *n* = 10 mice), perhaps because living longer they had more time to accumulate them. Only the ET-treated C26 hosts displayed lung nodules with a smaller prevalence and abundance per animal (6 mice out of 9, 67% of prevalence, with metastases number ranging from 0 to 7 per mouse). Despite displaying fewer metastases than the PM-treated group, the ET-C26 hosts lived less than the PM-C26 hosts, thus excluding lung metastases occurrence as cause of death.

Since PM01183 showed the best results on survival of cachectic mice, we chose to focus most of the next experiments on this compound.

### 2.2. Lurbinectedin Does Not Protect C26-Carrying Mice from Cardiac and Diaphragm Dysfunctions during Cancer Cachexia

Tumor burden, besides influencing body weight, also promotes cardiac and ventilatory dysfunctions, and tumor progression correlates with loss of diaphragm and cardiac functions in C26-mice [2,33].

We performed a wide-ranging analysis on the structure and/or function of cardiac and diaphragm tissues to verify whether PM01183 treatment prolonged survival of C26-mice by beneficially affecting these specific vital muscles. To have most of the C26-mice alive, we carried out these analyses on C26 hosts after 10–13 days from tumor implant, when their body weight loss was clear and the tumors had reached comparable sizes (Figure 1A,B). Within 13 days from the C26 implant, C26 hosts had received only 3–4 doses of 0.07 mg/kg PM01183, instead of the 9 doses in the first set of experiments (Figure 1). Using high-resolution ultrasounds, we confirmed that C26-mediated cachexia caused a reduction in diaphragm thickness by 67% (Figure 2A), diaphragm weight at sacrifice by about 25%, but not significantly, (Figure 2B) and diaphragm excursion by about 46% (Figure 2C). After PM01183 treatment, we found mild improvements in all the studied parameters of the diaphragm, although they did not differ significantly from vehicle-treated C26-mice (Figure 2A–C).

We also investigated the effect of PM01183 treatment on cardiac function using echocardiography. In vehicle-treated C26-mice, tumor progression led to a significant reduction of left ventricular ejection fraction (LVEF) by about 30% (Figure 3A), associated to reduction in right ventricle plus left ventricle weight, but not significantly (−10 and −7% respectively, Figure 3B). The reduction in LVEF may be in part due to the low heart rate induced by isoflurane anesthesia in both studied groups of C26-mice (Figure 3C). PM01183 treatment led to a significant improvement of heart rate by 30% that was not enough to significantly increase LVEF (+7% vs. C26 vehicle) (Figure 3A–C). Moreover, C26-mice had more myocardial injury, as indicated by more than doubled median concentration of circulating high-sensitivity cardiac troponin-T (hs-cTnT) (Figure 3D). In C26-mice treated with PM01183, we found 12% median lower concentration compared to the C26 vehicle group, even if it does not reach statistical significance (Figure 3D), suggesting that PM01183 may contribute mildly to restrain C26-induced heart injury. Lastly, we assessed collagen deposition levels using picrosirius red staining, but we did not find a significant increase of collagen in the hearts of vehicle-treated C26-mice compared with PBS-mice or with PM01183-treated C26-mice (Figure 3E).

Overall, these results suggest that 3–4 doses of PM01183 were not sufficient to exert any significant protection against diaphragm and cardiac dysfunctions. Maybe, PM01183 may prevent myocardial injury, as suggested by a mild lower plasma concentration of hs-cTnT, at least excluding an undesired cardiotoxic effect of this drug in vivo.

### 2.3. Lurbinectedin Does Not Protect either Fat or Hindlimb Muscles from Wasting but Restrains the Activation of the NF-κB/PAX7/Myogenin Axis in Tibialis Anterior of C26-Bearing Mice

Even though the main feature of cancer-induced cachexia is the loss of skeletal muscle mass, fat depletion may occur [1]. We used micro-computerized tomography (micro-CT), as we already did in [34], to measure the content of visceral fat tissue between L1 and L5 lumbar vertebrae in these mice. There was a reduction of about 40% in fat volume in C26-mice compared to healthy controls, and 3–4 doses of PM01183 did not prevent C26-induced fat wasting (Figure 4A).

Since mice injected with C26 adenocarcinoma cells already display muscle wasting 10–13 days from tumor implant [28,35], we tested whether 3–4 doses of 0.07 mg/kg PM01183 were sufficient to prevent cancer-induced muscle atrophy. At autopsy, in vehicle-treated C26-mice we confirmed reductions of about 12% and 13% in the muscle weights of the tibialis anterior (TA) and gastrocnemii, respectively, compared to PBS-mice (Figure 4B,C), while solei apparently were not reduced in C26-mice (Figure 4D). PM01183 did not significantly prevent these muscles from weight loss (Figure 4B–D). The mean cross-sectional area (CSA) of myofibers from gastrocnemii therefore showed a significant reduction comparing PBS-mice with vehicle-treated cachectic animals (C26 vehicle), and, in fact, the mean CSA dropped from 1730 μm^2^ to 1190 μm^2^ (Appendix A). This reduction was not restored by PM01183 (1260 μm^2^) (Appendix A). We then analyzed the mRNA expression of *atrogin-1* and *MuRF1*, the major ubiquitin ligases involved in protein degradation during cancer-mediated muscle atrophy [36,37]. As expected, both were significantly up-regulated in TA muscles from cachectic C26-mice, and PM01183 treatment failed to limit *atrogin-1* and *MuRF1* up-regulation (Figure 4E,F).

Pivotal mediators of cancer-induced cachexia are the transcription factors STAT3 and NF-κB, which, in turn, are mainly activated by the circulating pro-inflammatory cytokines, such as IL6 and TNFα, respectively [7,16,38]. Instead, C/EBP*β* acts as a myogenesis blocker during cancer cachexia and promotes the transcription of atrogin-1 [12,17]. As expected, western blot (WB) analysis showed that TA muscles from C26-mice had higher levels of phosphorylated NF-κB (pNF-κB), STAT3 (pSTAT3) and also C/EBP*β* protein content than healthy mice (Figure 4G–J). PM01183 did not inhibit activation of STAT3 (Figure 4G,I), but it strikingly blunted NF-κB activation (i.e., phosphorylation) (Figure 4G,H). The protein induction of C/EBP*β* is only partially, even if not statistically significant, reduced following PM01183 treatment (Figure 4G,J). We also measured the levels of PAX7, a transcription factor whose expression is sustained by NF-κB during C26-mediated cachexia [11], and those of myogenin, because the latter is known to induce atrogin-1 [39]. As expected, we found higher protein content of PAX7 and myogenin in C26-mice than PBS-mice (Figure 4G,K,L), as found by others [11,40]. Strikingly, PM01183 efficiently reduced PAX7 and myogenin levels (Figure 4G,K,L), but it is unlikely that such drug completely resolves the myogenic impairment typical of cachexia, as suggested by the failure of PM to restore C/EBP*β* to normal levels (Figure 4G,J).

These results indicate that PM01183 may contrast the myo-inflammation typical of cancer cachexia mediated by NF-κB, as suggested by the partial inhibition of the NF-κB/PAX7 axis in muscles. At least at this dose and time schedule, lurbinectedin appears to have limited effect on the enhanced protein catabolism due to activation of the STAT3/MuRF1/atrogin-1 axis and it is unable to obviate the myogenesis block due to activation of the C/EBP*β*-driven program, ultimately failing to fully protect muscle from wasting.

### 2.4. Lurbinectedin Abrogates Neither NF-κB Nor STAT3 Activation In Vitro in Atrophying Myotubes

To confirm and further dissect the possible effects of PM01183 on pro-inflammatory pathways directly on muscles, we analyzed the activation of NF-κB and STAT3 pathways in C2C12 myotubes exposed to IFNγ/TNFα and treated with PM01183 for 24 h.

We firstly assessed that PM01183 was not toxic in differentiated myotubes at doses of 0.5 and 1 nM but caused a reduction in cell viability by about 20% at doses ranging from 5 to 20 nM (Appendix A). We then found that 24 h treatment with 10 ng/mL IFNγ/TNFα reduced the myotube diameter by about 20% and observed that 1 nM PM01183 failed to prevent myotubes from IFNγ/TNFα-induced atrophy (Figure 5A,B). Consistent with in vivo data, we tested that the inductions of both *atrogin-1* and *MuRF1* in IFNγ/TNFα-treated myotubes were not prevented by PM01183 treatment (Figure 5C,D). Therefore, we investigated the NF-κB pathway and confirmed that it was highly induced by IFNγ/TNFα, but not inhibited by PM01183 treatment at all (Figure 5E,F). Conversely, we did not find an increased ratio of pSTAT3 over the total, in myotubes exposed to IFNγ/TNFα treatment, which is in accordance with the notion that STAT3 activation is preferentially induced by IL-6 [41] (Figure 5E,G).

We have repeated these in vitro experiments by better mimicking cachexia-like conditions in two distinct ways: by treating myotubes with medium conditioned by C26 cells (S-33%) or by co-culturing C2C12 with C26 cells by using Transwell devices (as in [42]) (TSW). We confirmed that both conditions were able to induce an evident atrophy in C2C12 myotubes, with no apparent toxicity, as seen by thinner differentiated cells (Appendix A), especially in those co-cultured with C26 cells (Appendix A). Contrary to what we found in C2C12 myotubes exposed to IFNγ/TNFα (Figure 5E–G), we assessed that both conditions were able to activate STAT3, but not NF-κB in myotubes (Appendix A). We also observed that under these atrophic conditions, 24 h treatment with 1nM PM01183 was not able to counteract atrophy in vitro nor to restrain STAT3 overactivation (Appendix A), further supporting in vivo findings, where PM01183 was unable to restrain STAT3 activation (Figure 4G,I).

Overall, these data indicate that PM01183 is able neither to attenuate atrophy nor to inhibit the activation of both the NF-κB and STAT3 pathways directly acting on muscle cells and suggest that the muscle inhibition of the NF-κB/PAX7 axis exerted by PM01183 in vivo might be related to changes in the pro-inflammatory milieu in C26-mice.

### 2.5. Lurbinectedin Inhibits C26-Induced Splenomegaly in Mice for as Long as the Treatment Lasts

Inflammation-mediated splenomegaly has been described in C26 tumor-bearing mice [30], and it may be caused by cancer-related immunosuppression in this cachectic animal model [43].

In the first set of experiments shown in Figure 1, we confirmed that C26-mice have spleens up to 15-fold bigger than healthy controls (Figure 6A,B) starting from about 20 days from tumor cell inoculation (Figure 6B), as already reported by Ju et al. [43]. Instead, C26-mice treated with PM01183, sacrificed at times when vehicle-treated C26-mice had splenomegaly, displayed spleens that were even smaller than PBS-mice (Figure 6A). Preliminary linear regression analysis suggested a delay of about 10 days in the onset of splenomegaly in C26-mice treated with PM01183, compared to vehicle-treated C26-mice (Figure 6B). More importantly, C26-induced splenomegaly was inhibited by PM01183 for as long as the treatment lasted (Figure 6B). In fact, by about ten days from the end of the treatment, PM01183-treated animals had spleens as abnormally big as those of the vehicle-treated C26-mice sacrificed 15 days earlier or less.

To further evaluate the effects on spleen exerted by PM01183, enlarged spleens of similar sizes from three C26-mice treated with vehicle or PM01183 (mean spleen weight: 1313 mg ± 219 for vehicle-treated C26-mice and 1307 mg ± 331 for PM01183-treated ones, unpaired t-test *p* = 0.988) were submitted to histopathological analysis (Appendix A). The same analysis was also performed on PBS-mice. Severe extramedullary hematopoiesis of the myeloid lineage was observed in enlarged spleens of C26-mice, regardless of the treatment (Appendix A). In the most severe cases, the normal structure of the spleen parenchyma was completely effaced by sheets of myeloid cells at different degrees of differentiation and rare residual lymphocytes, erythroid precursors and megakaryocytes. No major differences were found among vehicle- and PM01183-treated C26-mice at this time, perhaps because the effects of PM01183 were lost after 13–20 days from the termination of the treatment.

Since it has been reported that splenomegaly is directly linked to activation of a pro-inflammatory milieu [29], we asked whether the levels of circulating pro-inflammatory cytokines, also known to be induced during cancer cachexia in mice [5,6,44,45], were reduced by PM01183 in C26-mice sacrificed at 10–13 days from tumor injection. We found that plasma of C26 hosts had high levels of pro-inflammatory and/or pro-cachectic factors, such as IL-6 (Figure 6C), and a trend towards increased levels for M-CSF (Figure 6D) and IL-1β (Figure 6E) compared to controls (PBS). Unexpectedly, PM01183 failed to reduce the levels of these cytokines (Figure 6C–E). We also measured other cytokines that might be involved in splenomegaly during cancer cachexia, including granulocyte-macrophage colony-stimulating factor (GM-CSF), granulocyte-colony stimulating factor (G-CSF) and HMGB-1 (Appendix A) [46,47]. Again, we did not find any significant difference between vehicle-treated and PM01183-treated C26-mice and healthy controls, with the exception of a small tendency to reduced levels of HMGB-1 in PM01183-treated C26-mice that was not significant (Appendix A).

In summary, our results indicate that PM01183 may delay the onset of splenomegaly in C26-bearing mice, without affecting the circulating levels of pro-inflammatory cytokines analyzed, but in both treatment groups the increased spleen weight is not related to an early mortality, as clearly indicated by the regression curves (Figure 6B). Similar results were obtained at multi-organ level also in C26-mice subjected to ET743 treatment (Figure 7), including the beneficial effect against splenomegaly.

## 3. Discussion

Colon adenocarcinoma is one of the malignancies most likely to cause cachexia, a debilitating complication in advanced cancer patients that consists of depletion of different organs and tissues (e.g., peripheral skeletal muscles, heart, diaphragm), such as to compromise their function and ultimately lead to premature death [1,2,3]. Despite having been extensively tested as anti-cancer agents, the marine-derived alkaloids PM01183 and ET743 have never been evaluated against cachexia. Here, we found that PM01183 and ET743 substantially extended the life of C26 adenocarcinoma-bearing mice, by about 85% and 30%, respectively, with no effect on tumor growth and BW loss, like in patients where ET743 failed to limit advanced colorectal cancer [48].

The marked PM01183-induced increase in survival of C26-mice prompted us to perform a multi-organ analysis of these mice within 13 days from C26 implant when they had only received 3–4 out of 9 doses of PM01183, but when vehicle-treated C26 hosts were still alive and already displayed body wasting. Sophisticated methods were employed for the first time at once in this study to measure precisely the mass and function of several tissues: micro-CT was useful to evaluate abdominal fat tissue, as in [34], while ultrasound based-imaging was used to measure various cardiac (HR, LVEF and SF) and respiratory parameters (diaphragm thickness and excursion). PM01183-treated C26-mice, which received only 3–4 doses of drug, did not display significant amelioration in any of the tissues or organs analyzed but only slight improvements in various organs that altogether may presumably account for the overall wellness of the mice resulting in extended survival.

As shown in the radar scheme (Figure 7), wasting (and dysfunctions) of multiple tissues, such as hindlimb muscles, diaphragm, and abdominal fat were found in C26-mice already 10–13 days post injection. The most affected tissue at this time appears to be the diaphragm, whose thickness was reduced to almost 30% of controls but that was still compatible with life [2]. Thickness and excursion of the diaphragm were preserved by about 10–30% in PM01183-treated mice, although not significantly. Other muscles like those of the leg were slightly depleted, with soleus loss about 9%, but not significant, and gastrocnemius and TA loss about 14%, consistent with the notion that oxidative muscles are more resistant to cancer-induced wasting [49]. Conversely, the least affected tissue seemed to be the heart, whose weight was reduced in C26-mice at this time by only 6% of controls. Intriguingly, despite a slight reduction of its mass, the heart function was greatly reduced at this early time in C26-mice, displaying a significant reduction in SF% and LVEF in C26-mice. This apparently contrasts with data from Devine et al., who found decreased LVEF in C26-mice only 19 but not 14 days from tumor injections [50], but this may be because they used females known to be more resistant than males to C26-induced cardiac cachexia [51] or because of different ways to keep mice anesthetized.

Heart weights were mostly not affected by the treatment with both drugs. The lacking increase of collagen deposition even in vehicle-treated C26-mice with respect to controls is in contrast with that reported in Devine et al. [52]. This discrepancy may probably be due to the different C26 cells post-injection times of sacrifice between our study (10–13 days) and their work (24 days), indicating that, at earlier times, C26-mice display cancer cachexia associated to lower cardiac weight but not to higher cardiac fibrosis. We also evaluated cTNT levels because their change may represent an early cardiovascular event [53]. cTNT is a well-known biomarker of myocardial cell damage, predictive of adverse outcomes in patients with acute myocardial infarction, and heart failure [53,54,55]. Most interestingly, circulating cTNT has been found to be high also during age-related muscle wasting (i.e., sarcopenia) in various species [56], in plasma of some rodent models of cancer cachexia and in individuals suffering from non-small cell carcinoma of the lung [57,58]. We found higher levels of cTNT in C26-mice than PBS-injected ones. cTNT release may be under the control of pro-inflammatory cytokines, such as IL-6 and TNFα (both involved in C26-related cachexia), as also suggested by studies on how pharmacological blockade of these cytokines affects cTNT levels [59,60]. Thus, the partial but not significant reduction of cTNT levels in PM01183-treated mice may be due to some indirect anti-inflammatory effect of the drug in the rodent model of cachexia.

Conversely, PM01183 failed to limit atrophy in the hindlimb muscles (TA, gastrocnemius and soleus) analyzed, in terms of weight and CSA recovery in gastrocnemii. The compound was also unable to limit the C26-mediated activation of STAT3 in muscles and the C26-related induction of atrogenes, like *MuRF1* and *atrogin-1*. Instead, 3–4 doses of PM01183 partially blocked the C26-induced NF-κB/PAX7 axis in TA muscles, which may again support a specific anti-inflammatory property of this drug.

PAX7 is a typical transcription factor of muscle progenitor cells [61] and is overexpressed, through NF-κB signaling activation, in cachectic muscles of C26-mice [11]. Increased PAX7 expression sustains local inflammation in muscles of cachectic mice and underlies impaired muscle regeneration during cancer cachexia [62]. Moreover, PAX7 overexpression in hindlimb muscles is sufficient to induce atrophy, while tumor-bearing Pax7^+/−^ mice display less muscle atrophy than Pax7^+/+^ ones [11]. PM01183 attenuated the dysregulated activation of the NF-κB/PAX7 axis in muscle and also restrained myogenin levels, but this was not enough to contrast muscle atrophy, perhaps because the drug could not restrain the STAT3-mediated muscle degeneration. Overall, this suggests that PM01183 given in combination with STAT3 inhibitors could be useful to counteract cachexia in C26-mice, possibly resulting in even longer life extension of mice than treatment with only PM01183 or STAT3 inhibitors (i.e., sunitinib) [34].

Increased myogenin in muscles may participate in *atrogin-1* inductions, as it has been shown at least in denervation-induced atrophy [39], thus unravelling a dual role of myogenin in muscle physiology being detrimental or beneficial, depending on the pathological settings, as discussed in [39]. Interestingly, myogenin was found induced also in muscles upon castration-induced atrophy and following genetic ablation of androgen receptor activity in skeletal muscle by multiple means [63,64]. Puzzlingly, the expression level of myogenin in muscles of C26-bearing mice is controversial, since it appears to be reduced, unchanged or even increased in cachectic muscles [11,40,65,66]. Nonetheless, myogenin was found increased also in human cachectic muscle from patients with lung cancer [67] (where also an increased NF-Kb/PAX7 signaling occurred as in our samples).

C/EBP*β* is a transcription factor induced by inflammation (i.e., IL-1β) that blocks myogenic differentiation during cancer cachexia [12]. PM failure to properly restore basal C/EBP*β* levels in the muscles from C26 hosts indicates that PM is unable to efficiently promote a correct myogenic program. A precise myogenic program is a fundamental step to obviate muscle atrophy, as shown in a recent paper where treatment with IL-4, other than prolonging the life of C26-mice, as PM did, was also able to improve muscle mass by rescuing adult myogenesis [40]. Lack of efficacy of PM in reducing circulating levels of IL-1β may explain the concomitant inability of PM to effectively reduce the C/EBP*β* levels in cachectic muscles and thus to ameliorate muscle atrophy in C26 hosts. Despite this evidence, other studies are needed to discern how PM interferes with myogenesis, such as evaluating the regeneration process upon cardiotoxin-induced muscle injury in C26 hosts, in the presence or not of PM treatment, as done by Costamagna et al. for IL-4 [40].

Since we could not exclude a local effect of PM01183 on muscles, we tested whether it was able to directly inhibit NF-κB activation in isolated atrophying myotubes. We found that IFNγ/TNFα-mediated activation of the NF-κB pathway was not inhibited by PM01183, which was also unable to preserve myotube diameters and to restrain *atrogin-1* and *MuRF1* inductions. Taken together, our data show no overall anti-atrophic effect of PM01183 but confirm that this drug interferes with the NF-κB pathway also in cachectic muscles, in addition to tumoral cells [68]. Despite muscle activation of the NF-κB axis *per se* is known to cause MuRF1 induction and muscle atrophy [7], its de-activation exerted by PM01183 was unable to revert the atrophic process in muscle of cachectic mice. Our data may contrast with the report by Wysong and collaborators, showing that NF-κB inhibition by a specific peptide protects the body and muscle from wasting in C26-mice [69]. Surprisingly, the authors did not measure either tumor sizes over time or the effects of the compound on mouse survival and splenomegaly, making the comparison with PM01183 more difficult.

Since our results indicate lack of a local and direct effect of PM01183 on muscles, systemic activity of this drug on the pro-inflammatory milieu, already demonstrated for its natural lead compound [70], is plausible. Among all the tissues analyzed in C26-mice, PM01183 had the strongest effect only on the spleen. Our data showed that PM01183 fully prevented C26-induced splenomegaly in mice. In agreement with these findings, similar results were obtained for ET743. Splenomegaly onset results from a dysfunctional immune reaction [71] and is a marker of tumor progression [72] and a common feature of C26-bearing mice [29,30]. The spleen is the main site of myeloid cell differentiation which, in oncogenesis, can differentiate into myeloid-derived suppressor cells (MDSCs), a mixed population of immature myeloid cells that critically affect the host metabolism [29] and immune system [73]. Expansion of splenic MDSCs has been linked with cachexia, the acute phase response with dysregulated production of pro-inflammatory mediators and reduced survival [29,74,75]. Thus, drugs to deplete MDSCs may offer a novel strategy for limiting cachexia-related systemic inflammation and premature death. Consistently, splenectomy in mice before C26 tumor injection resulted in extension of their survival [76,77] with no effect on cachexia, similarly to the PM01183-treated C26-mice. Our results may therefore suggest that the longer survival of C26-mice treated with PM01183 might be possibly related to its inhibitory effect on spleen-derived MDSCs (as indirectly seen by reduced spleen sizes), already demonstrated both in vitro and in vivo [78]. Notably, lurbinectedin may have effects on other cancer-related mechanisms involving tumor-associated macrophages or on angiogenesis that may account for the increased survival and that have not been investigated in this study [79,80]. Overall, the exact mechanisms linking splenomegaly, MDSC expansion and possible longer survival during cancer cachexia require further research.

Lastly, we asked whether the anti-splenomegaly effect of PM01183 could have involved some systemic reduction of pro-inflammatory cytokines, resulting in the drug’s beneficial effects on circulating cTNT levels and NF-κB/PAX7 axis activation in muscles. We confirmed the high plasma levels of pro-inflammatory mediators such as IL-6, M-CSF and IL-1β in C26-mice, but PM01183 did not affect the plasma concentration of these cytokines. We also measured plasma levels of GM-CSF, G-CSF, HMGB-1 and pentraxin 3, because they have all been linked to splenomegaly [46,47,81]. Unfortunately, we did not observe any effects of PM01183 that could explain the delay in splenomegaly, with the exception of a tendency to reduced HMGB-1 levels in PM-treated C26-mice that was not significant. This may be consistent to what was found by Soda et al., showing that splenectomized C26-mice live longer, with no changes in their serum levels of TNFα or IL-6 [76].

The systemic effects of PM01183 partially mimic those reported by Nissinen et al. [82], showing that blocking the activin receptor type 2 (ACVR2B) ligands efficiently attenuated cachexia, improved survival and reduced splenomegaly in C26-mice, but this was not explained by fewer markers of MDSCs nor by restrained levels of IL-6 or IL-1β. If so, contrary to the findings reported in [78], PM01183 could be able to reduce spleen size without affecting the activation/proliferation of MDSCs and pro-inflammatory cytokine production. Therefore, the lack of efficacy of PM01183 in reducing the circulating inflammatory cytokines analyzed suggests that PM01183-mediated NF-κB/PAX7 axis inhibition and cTNT levels reduction may be due (i) to inhibition of other pro-inflammatory mediators not investigated or (ii) to a novel effect of this drug on other tissues that have not been analyzed (i.e., liver). Future investigation with long-term treatment of PM01183 may be needed to unravel the mechanism underlying its anti-splenomegaly ability and the increased survival.

## 4. Materials and Methods

### 4.1. Cell Culture

C2C12 (ATCC, Manassas, VA, USA), a myoblast cell line from the C3H mouse strain, was grown in DMEM (Dulbecco’s modified Eagle’s medium, Gibco, Waltham, MA, USA), supplemented with fetal bovine serum (FBS) (Euroclone, Pero, Italy) and 2 mM L-glutamine, and maintained in culture at 37 °C with 5% CO_2_. Myoblasts were differentiated into myotubes when they reached 80% confluence and were cultured for four days in DMEM and 2 mM L-glutamine (BioWest, Nuaillè, France), supplemented with horse serum (HS) (Euroclone, Pero, Italy), at 37 °C and 8% CO_2_. The differentiation medium was changed every two days. To investigate the effect of PM01183 on NF-κB pathway activation, myotubes were treated on the fourth day of differentiation for 24 h with 1 nM PM01183 or IFNγ (10 ng/mL)/TNFα (10 ng/mL) or in combination, using DMSO as vehicle.

C26 is a colorectal adenocarcinoma cell line from BALB/c mice, grown in DMEM supplemented with 10% FBS and 2 mM L-glutamine at 37 °C with 5% CO_2_. These cells were kindly donated by Prof. Mario Paolo Colombo (IRCCS-Istituto Nazionale dei Tumori, Milan, Italy) and authenticated as in [83]. The cells were not contaminated by mycoplasma.

### 4.2. Drugs

Trabectedin (Yondelis) and lurbinectedin were kindly provided by PharmaMar (Madrid, Spain). They were dissolved in sterile water and further diluted in saline immediately before use.

### 4.3. Mice and Tumor Model

C26 cells were seeded at a density of 17,000 cells/cm^2^ and 48 h later injected subcutaneously (1 × 10^6^ cells) into the upper right flank of 10-week-old male BALB/c mice (BW 22–25 g) (Harlan Laboratories, Lesmo, Italy). Mice were weighed the day of the injection, then every two days until they began to lose weight, after which they were weighed daily. Body weights, tumor growth and food intake were recorded as detailed in [28]. In particular, the food intake was measured on a per-cage basis. The mean cumulative food intake (*n* = 2 cages) shown in Figure 1D is the sum over time of all the food consumed on average by the animals during the whole experimental period, as done by others [84,85]. The length and width of tumors were measured using digital calipers, and tumor volume was assessed using the formula as follows: volume (mm^3^) = (W^2^) × L/2 (W = width; L = length). The number of lung metastases were counted from animals of each group in the survival experiment shown in Figure 1. In detail, lungs were excised and fixed in Bouin’s solution (Bio-Optica, Milan, Italy) and superficial metastatic nodules were counted and measured using a dissecting microscope, as described in [86]. In the first set of experiments (Figure 1), from day 3 to day 21, C26-mice were randomized to receive into their tail vein, three times a week for three weeks, ET743 or PM01183, 0.05 mg/kg and 0.07 mg/kg, respectively, and sacrificed when they had clear signs of distress. These mice were used for Figure 1 and for Figure 6A,B. The group of mice indicated as "C26 vehicle" was plotted until four mice survived in Figure 1A,B. In the second set of experiments, C26-mice received only 3–4 doses of PM01183 at 0.07 mg/kg, instead of the 9 doses of the first set of experiments, and they were sacrificed 10–13 days after tumor injection. In accordance with institutional guidelines, animals were killed when at least four out of five signs of distress (loss of mobility, kyphosis, ruffled fur, dehydration, tremor) were present or when more than 20% of BW was lost in 72 h. All the in vivo experiments were carried out in a blinded setting.

Procedures involving animals and their care were conducted in conformity with institutional guidelines in compliance with national and international laws and policies. The Mario Negri Institute for Pharmacological Research IRCCS (IRFMN) adheres to the principles set out in the following laws, regulations and policies governing the care and use of laboratory animals: Italian Governing Law (D.lgs 26/2014; Authorization no. 19/2008—A issued 6 March 2008 by Ministry of Health); Mario Negri Institutional Regulations and Policies providing internal authorization for persons conducting animal experiments (Quality Management System Certificate—UNI EN ISO 9001:2015—Reg. no. 6121); the National Institutes of Health (NIH) Guide for the Care and Use of Laboratory Animals (2011 edition) and European Union (EU) directives and guidelines (European Economic Community (EEC) Council Directive 2010/63/UE).

### 4.4. Diaphragmatic Ultrasonography

Echographic diaphragmatic evaluation was performed in mice in the supine position. The 30 MHz mechanical probe was placed over one of the lower intercostal spaces in the right anterior axillary line for the right diaphragm, and the hepatic acoustic window was used [87]. The ultra-sound beam was directed to the hemi diaphragmatic domes. During inspiration, the normal diaphragm contracts and moves caudally toward the transducer; an echocardiographic M-mode tracing and a 2-D cineloop (including at least five respiratory cycles) were recorded to measure the diaphragmatic excursion from the baseline (i.e., at the end of normal expiration) to the end of eupneic respiration as well (i.e., after the inhalation of a tidal volume). Diaphragmatic thickness was also assessed and measured at the end of exhalation during eupneic respiration (i.e., at lung volume of functional residual capacity). Five measurements were recorded and averaged. The whole examination was accomplished in five minutes.

### 4.5. Echocardiography

#### 4.5.1. Image Acquisition

Echocardiography was done with a 30 MHz mechanical probe (VisualSonics, Vevo 770, Toronto, ON, Canada) on mice anesthetized with isoflurane (0.5–1.5% in O_2_). Animals were positioned on a rail system for maintenance of the body temperature (37 °C ± 0.5 °C) and the probe positioned under electrocardiographic (EKG) and respiratory monitoring for the entire duration of the examination.

First, a parasternal long-axis (pLAX) B-mode image was acquired, optimizing the LV length for LV volume measurements and calculations. Single frames and cine loops containing five cardiac cycles (25–35 frames per cardiac cycle at a frame rate of 400–500/s when possible) were stored in DICOM format and measurements were taken offline [88].

#### 4.5.2. LV Volumes and LVEF Calculation

Measurements were made offline by two sonographers (I.R., L.S.) blinded to experimental groups. LV endocardial areas were drawn manually from end-diastolic and end-systolic frames from LAX view. LV end-diastolic and end-systolic volumes (LVEDV and LVESV) were calculated as single-plane modified Simpson (ComPACS software, Medimatic S.R.L, Genoa, Italy) from the pLAX view.

### 4.6. Micro-CT Analysis

In the second set of experiments, mice were anesthetized with a continuous flow of 3% isoflurane/oxygen mixture and positioned prone with both legs at right angles. The region spanning the entire torso to the distal tibia of each mouse was scanned with an Explore Locus micro-CT scanner (GE Healthcare, Chicago, IL, USA) without contrast agents. Four hundred micro-CT projections of the animals were acquired over 360° using 80 kV, 450 μA current and 100 ms of acquisition time with a resolution of 93 μm. The reconstructed 3D images were visualized and analyzed using MicroView analysis software (GE Healthcare, Chicago, IL, USA). The amount of adipose tissue was determined as in [89]. The gray-scale histogram of the reconstructed images presents a peak that indicates the presence of fat. One low and one high gray-scale threshold corresponding to that peak were chosen. The fat was quantified as the sum of the volumes of all the voxels with a gray-scale value between the low and the high thresholds. The analysis was done on the abdominal region (between the proximal end of lumbar vertebrae L1 and L5).

### 4.7. Collagen Detection in Cardiac Tissues by Picrosirius Red Staining

After euthanasia, hearts were excised from mice, with careful dissection from surrounding tissues. The left ventricle with the septum and the right ventricle were separated from the atria, weighed, fixed by immersion in 10% buffered formalin and embedded in paraffin.

Nine-μm thick sections of paraffin-embedded ventricles were used for picrosirius red staining to measure interstitial collagen in blind conditions. Sections were dewaxed for 40 min at 70 °C and 10 min in xylene, 5 min in ethanol 100%, 2 min in ethanol 100%, 2 min in ethanol 96%, 2 min in ethanol 70% and a brief wash in distilled water. Then, samples were stained with 0.1% picrosirius red for 1 h and then washed in distilled water for 2 min and 0.01 M HCl, pH 2, for 2 min. Finally, sections were dehydrated for 30 s in 96% ethanol, 1 min in 100% ethanol, 5 min in 100% ethanol and 10 min in xylene. Pictures were acquired with a Virtual Slide Microscope VS120 (40× magnification, Olympus, Shinjuku, Japan). Collagen was measured excluding vessels from analysis, using ImageJ software (National Institutes of Health, Bethesda, MA, USA).

### 4.8. RNA Isolation from Cultured Cells or Muscles and Reverse Transcription

Total RNA was isolated from cells or muscles with QIAzol Lysis Reagent (Qiagen, Hilden, Germany) and a miRNeasy Kit (Qiagen, Hilden, Germany). RNA concentration, purity and integrity were measured in a spectrophotometer (NANODROP 1000, ThermoFisher Scientific, Waltham, MA, USA). Cells from one well of a six-well plate were resuspended in 700 μL of QIAzol. The frozen muscle of interest was cut perpendicular to the tendon and supplemented with 700 μL of QIAzol Lysis Reagent (Qiagen, Hilden, Germany). The tissue was then lysed with a T25 digital Ultra-Turrax homogenizer (IKA, Staufen, Germany). From this point onwards, the extraction procedure was identical to that for RNA extraction from cells. A High-Capacity cDNA Reverse Transcription Kit (Applied Biosystems, Waltham, MA, USA) was used for the reverse transcription of RNA to cDNA. Transcription was done for 1 µg of RNA in 40 μL, according to the following reaction schedule: 25 °C for 10 min; 37 °C for 2 h; 85 °C for 5 min. Samples were then stored at −20 °C.

### 4.9. Quantitative Real-Time Polymerase Chain Reaction (PCR)

Total mRNA was analyzed using TaqMan Mix (ThermoFisher Scientific, Waltham, MA, USA) or the fluorescent intercalating DNA SYBR Green (Qiagen, Hilden, Germany). *IPO8* (*Importin 8*) and *TBP* (*TATA binding protein*) were used as housekeeping genes. We loaded each well of 96-well plates with 20 ng of cDNA (2 μL) obtained with reverse transcription, supplemented with either the primers and the SYBR Green mix (Qiagen, Hilden, Germany) or the probe and the TaqMan Mix (ThermoFisher Scientific, Waltham, MA, USA). In both cases, water was added to a volume of 11 μL. The PCR cycle for real-time PCR was as follows: step 1, 95 °C for 15 min; step 2, 95 °C for 25 s; step 3, 60 °C for 1 min; repeating steps 2 and 3 for 40 cycles. We used a 7900HT Fast Real-Time PCR System (ThermoFisher Scientific, Waltham, MA, USA).

### 4.10. Protein Extraction and Western Blot

Total proteins were extracted from myotubes using radioimmunoprecipitation assay (RIPA) buffer, with the final addition of 4% sodium dodecyl sulphate (SDS) and phosphatase/protease inhibitors (Roche, Basel, Switzerland). The frozen muscle of interest was cut perpendicular to the tendon, supplemented with 20 μL/mg of RIPA buffer, enriched as indicated above and lysed with a T25 digital Ultra-Turrax homogenizer (IKA, Staufen, Germany). The final protein concentration was determined using bicinchoninic acid assay (BCA, Pierce, Waltham, MA, USA). Then, 10–40 µg of proteins were added to 4× Laemmli sample buffer (Biorad, Hercules, CA, USA), previously mixed with 10% β-mercaptoethanol (Sigma, St. Louis, MO, USA), and boiled at 97 °C for 7 min.

Proteins were separated using electrophoresis on 4–20% sodium dodecyl sulphate-polyacrylamide gel electrophoresis (SDS-PAGE) (Biorad, Hercules, CA, USA) and transferred to a polyvinylidene difluoride membrane (GE Healthcare, Chicago, IL, USA) that was then saturated for 2 h at room temperature in a solution of 5% bovine serum albumin (BSA) or milk in a buffer of 20 mM Tris, 150 mM NaCl and 0.1% Tween-20 (Sigma, St. Louis, MO, USA) (TBS-T buffer).

The membrane was then incubated with the primary antibody O/N at 4 °C. The following primary antibodies were used: 1:10,000 anti-vinculin (V9264, Sigma, St. Louis, MO, USA), 1:1000 anti-phosphoSer536-NF-κB p65 (3033, Cell Signaling, Danvers, MA, USA), 1:1000 anti-NF-κB p65 (8242, Cell Signaling, Danvers, MA, USA), 1:1000 anti-phosphoTyr705-STAT3 (9145, Cell Signaling, Danvers, MA, USA), 1:1000 anti-STAT3 (9139, Cell Signaling, Danvers, MA, USA), 1:200 anti-PAX7 (Hybridoma Bank, Iowa City, IA, USA), 1:200 anti-C/EBP*β* (sc-7962, Santa Cruz Biotechnology, Dallas, TX, USA), 1:200 anti-myogenin (Hybridoma Bank, Iowa City, IA, USA). After overnight incubation with the primary antibody, the membrane was washed for 30 min with TBS-T buffer and incubated for 1 h at room temperature with the secondary antibody, diluted in a solution of 1% BSA or 1% milk in TBS-T buffer. The membranes were then washed for 30 min in TBS-T buffer to remove the excess of unbound antibody. Secondary antibodies were conjugated to alkaline phosphatase (Promega, Madison, WI, USA) and detected with CDP-star substrate (ThermoFisher Scientific, Waltham, MA, USA). Band intensities were analyzed using ImageJ software (National Institutes of Health, Bethesda, MA, USA).

### 4.11. Cardiac Troponin and Cytokine Measurements

Plasma was collected and immediately added to 10% 0.5 M EDTA as an anticoagulant, centrifuged for 10 min at 10,000 rpm at 4 °C, and then stored at −80 °C. Hs-cTnT was measured with an electrochemiluminescence assay (Cobas, Roche Diagnostics, Rotkreuz, CH, Switzerland). The detection range is 3–10,000 ng/L. IL-6 and IL-1β levels in murine plasma were measured using MILLIPLEX MAGPMAG-24K-1 from Merck Millipore (Burlington, MA, USA). M-CSF levels in plasma were measured with MILLIPLEX MCYTOMAG-70K-1 from Merck Millipore (Burlington, MA, USA). The detection ranges are as follows: for IL-6, 7–5000 pg/mL, for IL-1β, 14–10,000 pg/mL, for M-CSF, 3.2–10,000 pg/mL. Values below the detection limit were not included in the analyses and related graphs. This technology allows multiplexed analysis of several analytes in one sample, exploiting fluorescent magnetic beads coated with specific antibodies.

### 4.12. Myotube Diameter Measurements

Myotube diameters were measured on the fourth day of differentiation in blind conditions by at least two operators using ImageJ software (National Institutes of Health, Bethesda, MA, USA). Pictures of myotubes were acquired with an Olympus Microscope IX71 (10× magnification, 10× ocular lens, Olympus, Shinjuku, Japan) with Cell F (2.6 Build1210, Olympus, Shinjuku, Japan) imaging software for Life Science microscopy (Olympus Soft Imaging solution GmbH, Munster, Germany).

### 4.13. Statistical Analysis

Sample size was determined by power analysis with G*Power on the basis of similar experiments previously published by our laboratory. For statistical analysis, data (means  ±  standard errors of the mean, SEMs) were analyzed with GraphPad Prism 7.8 for Windows (Graph-Pad Software, San Diego, CA, USA) and Statview Software for Windows (SAS StatView for Windows, Redmond, WA, USA), with the following statistical tests: log-rank (Mantel‒Cox) test; linear regression *t*-test; ordinary one-way analysis of variance (ANOVA) for multiple comparisons followed by Tukey’s post-hoc test or Kruskal‒Wallis with post-hoc Dunn’s multiple comparison test. *p*-values ≤ 0.05 were considered significant.

## 5. Conclusions

Overall, our results clearly indicate that PM01183 does not protect C26-mice from cancer-induced cachexia but did evidently increase their survival. The restrained splenomegaly exerted by lurbinectedin cannot fully explain the increased survival of C26-bearing mice, because some die with normally sized spleen, regardless of treatments. Future studies will also aim at dissecting how PM01183 exerts such effect.

## Figures and Tables

**Figure 1 cancers-12-02312-f001:**
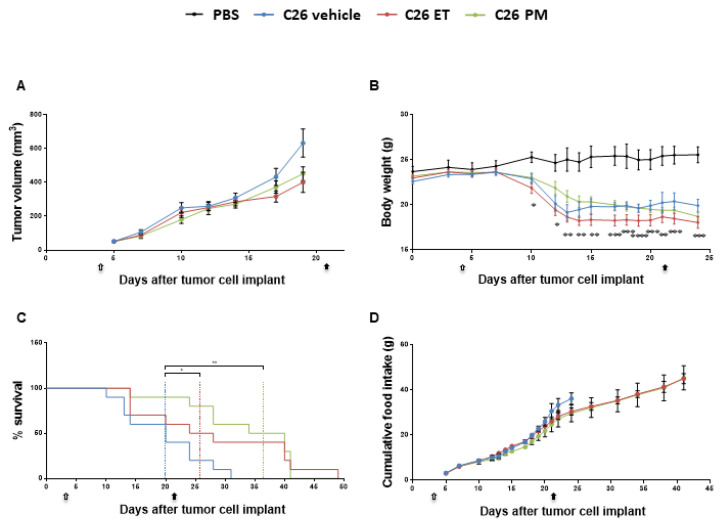
**PM01183 and ET743 extend survival of C26-bearing mice, with no anti-tumor effects.** C26-mice (10/group) were randomized to receive ET743 (ET, 0.05 mg/Kg) or PM01183 (PM, 0.07 mg/Kg), three times a week for three weeks, or vehicle for the time indicated (arrows indicate the start and end of treatment, from day 3 to day 21). Tumor volumes were measured manually with a caliper and are shown in (**A**). Body weights are plotted over time (**B**). Mice were euthanized when they lost 20% of their body weight over 72 h and/or showed signs of distress; survival curves are depicted (**C**). Cumulative food intake is shown (**D**). PBS-treated mice (*n* = 3) were used as controls. Results are plotted as mean ± SEM (**A**,**B**,**D**). * *p* ≤ 0.05, ** *p* ≤ 0.01, *** *p* ≤ 0.001 and **** *p* ≤ 0.0001 for PBS vs. C26 vehicle, one-way ANOVA with post-hoc Tukey’s multiple comparison test (**A**,**B**,**D**) or log-rank (Mantel‒Cox) (**C**).

**Figure 2 cancers-12-02312-f002:**
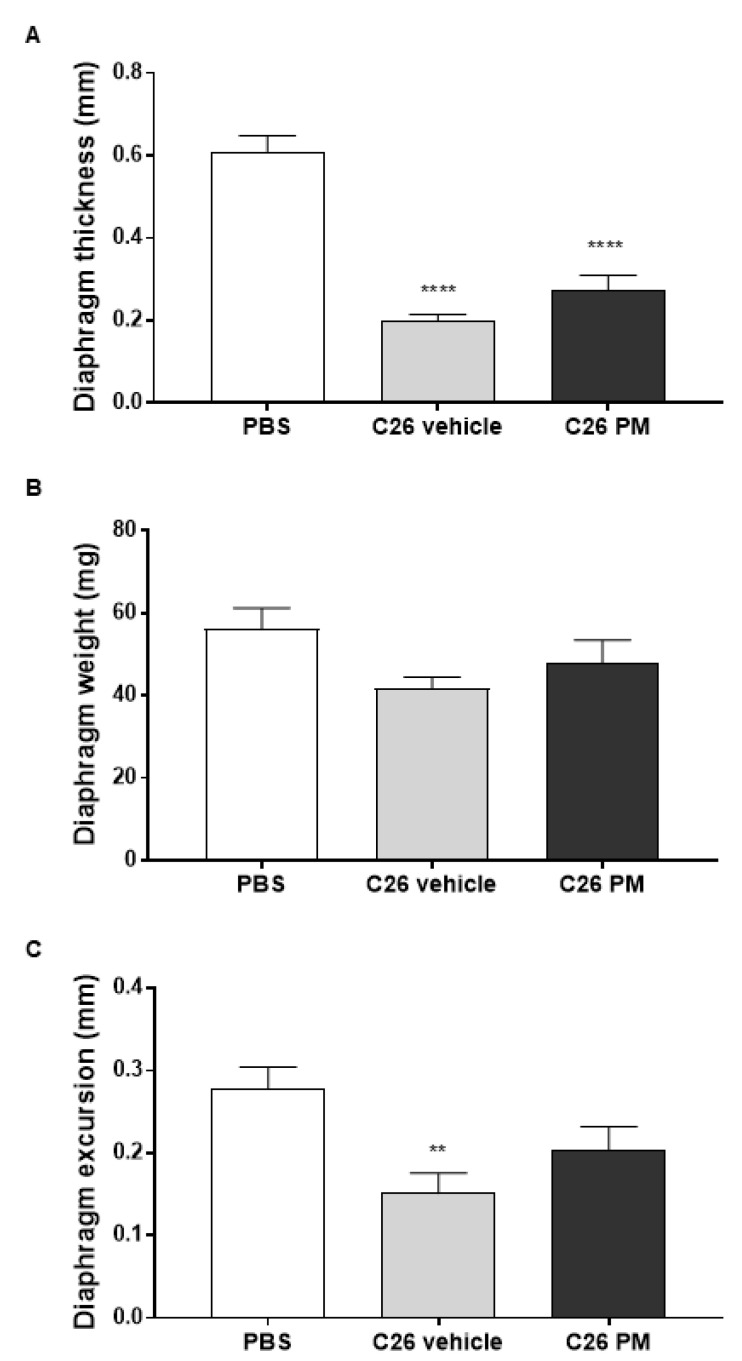
**PM01183 does not protect the diaphragm of C26-bearing mice from wasting and dysfunction.** C26-mice (10/group) were randomized to receive PM01183 (0.07 mg/Kg) or vehicle and euthanized after 3–4 doses, 10–13 days from the C26 implant. Diaphragm thickness (**A**) and excursion (**C**) were measured before death using ultrasound-based imaging. Diaphragm weights measured at sacrifice are also shown (**B**). PBS-treated mice (*n* = 10) were used as controls. All results are plotted as mean ± SEM. ** *p* ≤ 0.01, **** *p* ≤ 0.0001, one-way ANOVA with post-hoc Tukey’s multiple comparison test (**A**,**B**) or Kruskal‒Wallis with post hoc Dunn’s multiple comparison test (**C**).

**Figure 3 cancers-12-02312-f003:**
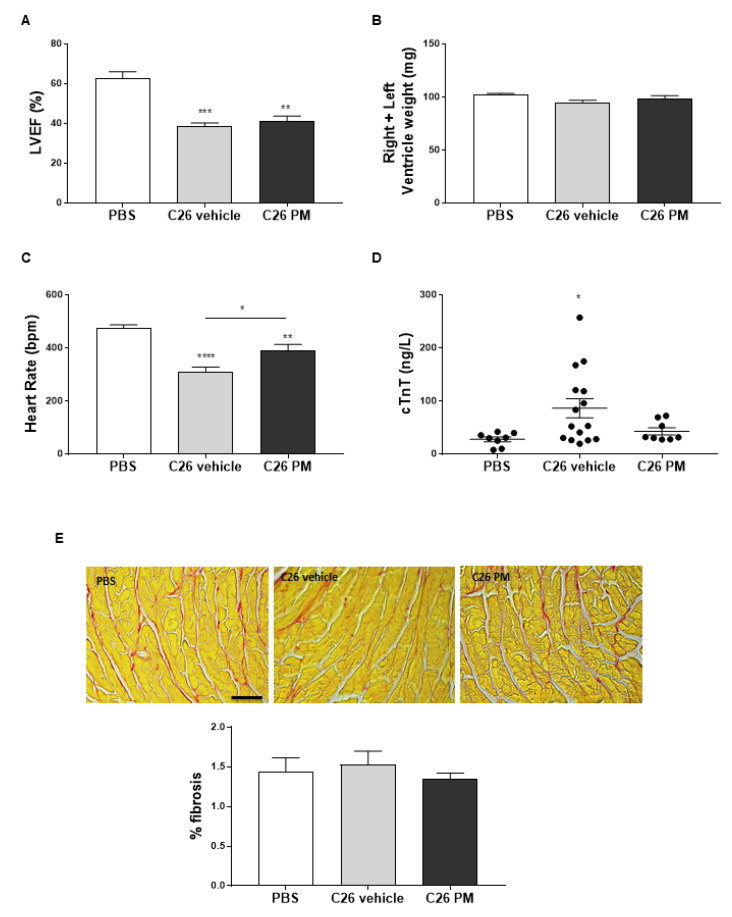
**PM01183 does not protect hearts of C26-bearing mice from wasting or dysfunction.** C26-mice (10/group) were randomized to receive PM01183 (0.07 mg/Kg) or vehicle and euthanized after 3–4 doses, 10–13 days from the C26 implant. Left ventricle ejection fraction (LVEF) (**A**) and heart rate (**C**) were measured using ultrasound-based imaging. Right plus left ventricle weights are shown in (**B**). High sensitivity cardiac troponin-T (hs-cTnT) plasma levels were measured using ELISA (**D**). Collagen deposition in 9 μm-thick sections of paraffin-embedded heart was determined using picrosirius red staining (upper panel, scale bar: 50 μm) and quantified as a percentage of the total area using ImageJ software (lower panel) (**E**). PBS-treated mice (*n* = 10) were used as controls. All results are plotted as mean ± SEM. * *p* ≤ 0.05, ** *p* ≤ 0.01, *** *p* ≤ 0.001, **** *p* ≤ 0.0001, Kruskal‒Wallis with post hoc Dunn’s multiple comparison test (**A**) or one-way ANOVA with post-hoc Tukey’s multiple comparison test (**B**–**E**).

**Figure 4 cancers-12-02312-f004:**
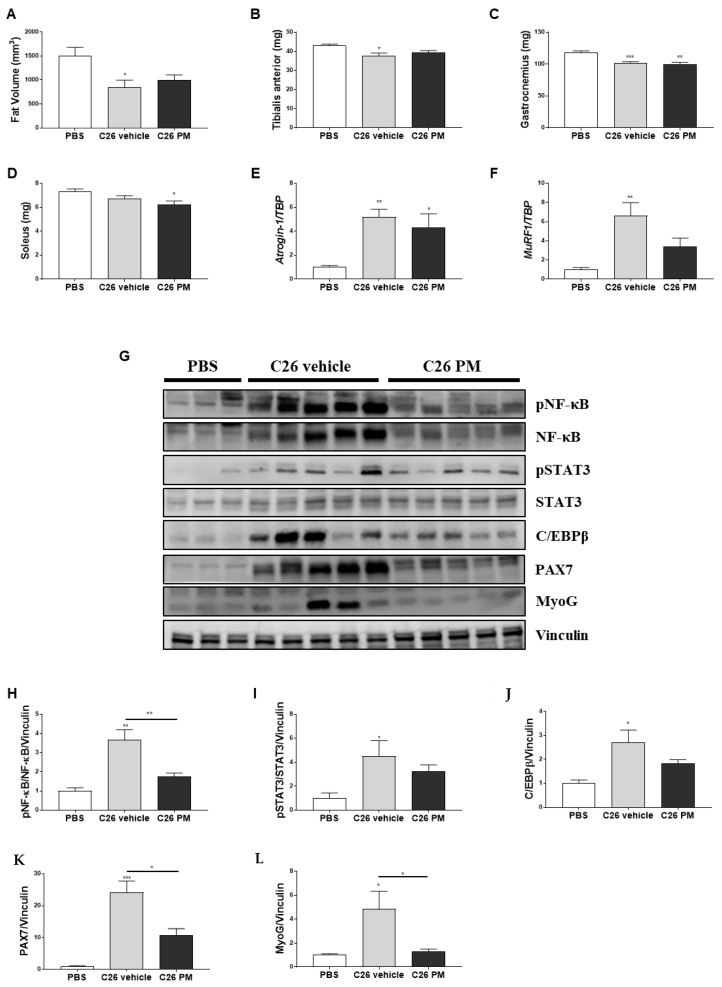
**PM01183 does not prevent fat wasting and hindlimb muscle atrophy but restrains NF-κB/PAX7 axis activation in tibialis anterior muscles of C26-bearing mice.** C26-mice (10/group) were randomized to receive PM01183 (0.07 mg/Kg) or vehicle and euthanized after 3–4 doses, 10–13 days from the C26 implant. PM01183 does not reverse fat tissue depletion in C26-bearing mice. Fat volume was quantified using micro-CT scanning from L1 to L5 (**A**). Tibialis anterior, TA, (**B**), gastrocnemii (**C**) and solei (**D**) muscles of PBS-mice and vehicle- or PM01183-treated C26-mice (after 10–13 days of treatment, 10/group) were weighed. TA muscles were analyzed using qPCR for *atrogin-1* (**E**) and *MuRF1* expression (**F**), *n* = 8–13. TA muscles were analyzed using western blot for pNF-κB/NF-κB, pSTAT3/STAT3, PAX7, C/EBP*β* and myogenin (MyoG) (**G**) and the related band quantitation is plotted, *n* = 3–5 (**H**–**L**). *Ipo8* and vinculin were used as housekeeping gene and internal loading control in qPCR and western blot, respectively. PBS-treated mice were used as controls. All results are plotted as mean ± SEM. * *p* ≤ 0.05, ** *p* ≤ 0.01, *** *p* ≤ 0.001, one-way ANOVA with post-hoc Tukey’s multiple comparison test (**A**–**E**,**H**–**L**) or Kruskal‒Wallis with post hoc Dunn’s multiple comparison test (**F**). The whole western blot images are shown in Appendix A.

**Figure 5 cancers-12-02312-f005:**
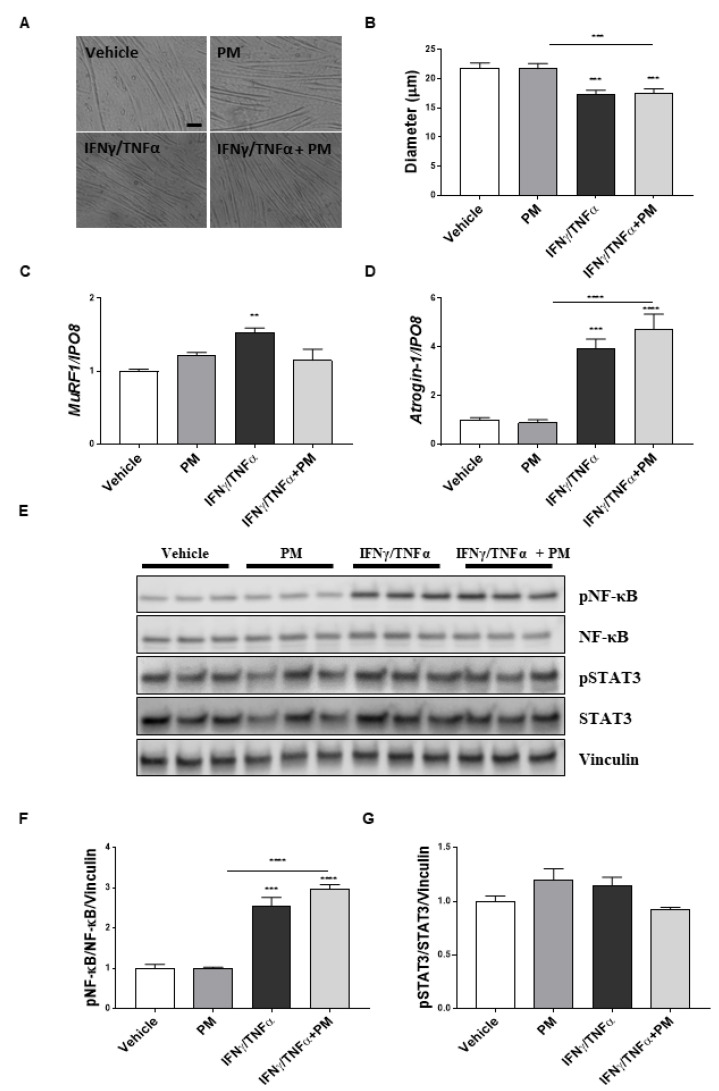
**PM01183 does not directly prevent NF-κB activation in atrophying myotubes.** Representative images of myotubes treated with vehicle or 1 nM PM01183 or 10 ng/mL IFNγ/10 ng/mL TNFα or in combination for 24 h are shown (scale bar: 100 μm) (**A**). Myotube diameters were measured manually with ImageJ software, *n* = 87 (**B**). *MuRF1* (**C**) and *atrogin-1* (**D**) expression levels were evaluated using qPCR, *n* = 4. *Ipo8* was used as housekeeping gene. Western blot analysis for pNF-κB/NF-κB and pSTAT3/STAT3 is shown in (**E**) and the related band quantitation is plotted, *n* = 3 (**F** and **G**). Vinculin was used as loading control. All results are plotted as mean ± SEM, ** *p* ≤ 0.01, *** *p* ≤ 0.001, **** *p* ≤ 0.0001; one-way ANOVA with post-hoc Tukey’s multiple comparison test (**B**,**D**,**F**,**G**) or Kruskal‒Wallis with post-hoc Dunn’s multiple comparison test (**C**). The whole western blot images are shown in Appendix A.

**Figure 6 cancers-12-02312-f006:**
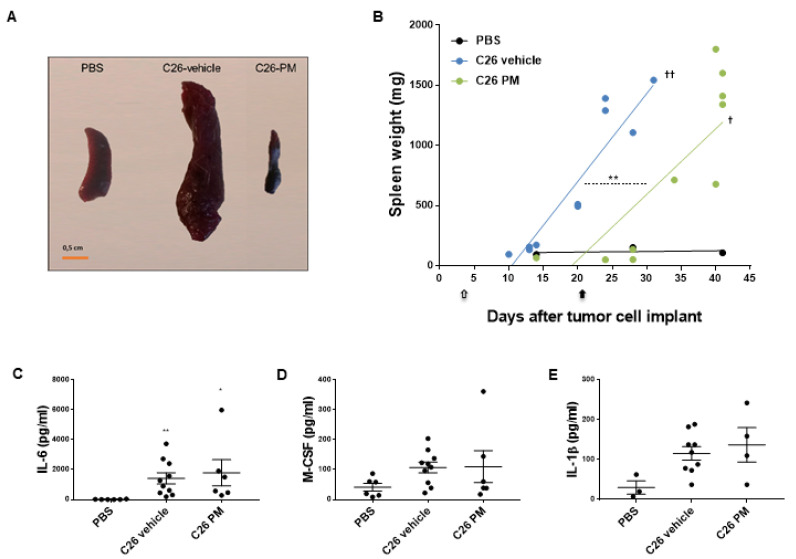
**PM01183 inhibits C26-induced splenomegaly.** Splenomegaly was visually inspected (**A**, scale bar 0.5 cm) 28 days after tumor injection. Spleen weights from mice euthanized when showing signs of distress are shown (**B**). Arrows indicate the length of treatments. Plasma levels of IL-6 (**C**), M-CSF (**D**) and IL-1β (**E**) were measured using multiplex assays in PBS-mice (6/group), C26-mice treated with PM01183 (6/group) or vehicle (10/group) after 10–13 days from tumor injection. PBS-treated mice were used as controls. Results are plotted as mean ± SEM. Linear regression t-test. Intercepts: ** *p* ≤ 0.01 (C26-vehicle vs. C26-PM); slopes: †† *p* ≤ 0.01 (C26-vehicle vs. PBS), † *p* ≤ 0.05 (C26-PM vs. PBS) (B). * *p* ≤ 0.05, ** *p* ≤ 0.01, Kruskal‒Wallis with post-hoc Dunn’s multiple comparison test (**C**,**D**) or one-way ANOVA with post-hoc Tukey’s multiple comparison test (**E**).

**Figure 7 cancers-12-02312-f007:**
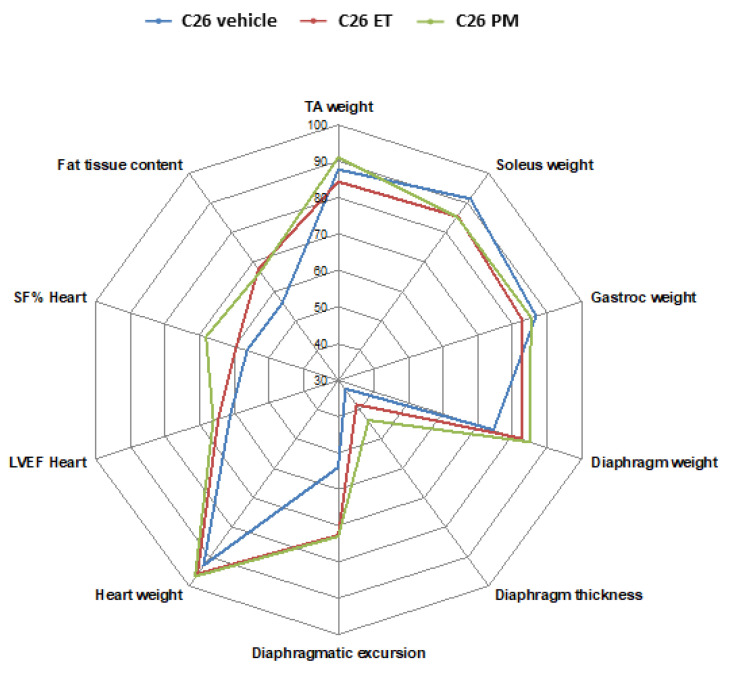
**In 10–13 days from injection, C26 growth in mice causes atrophy and dysfunction of multiple tissues that are not fully reversed by ET743 or PM01183**. PBS-injected mice were used to measure multiple parameters and plotted as 100% at the extremity of the scheme. Weights of five muscles (TA, soleus, gastrocnemius indicated as gastroc, diaphragm and heart), thickness and excursion of the diaphragm, shortening fraction (SF) and left ventricular ejection fraction (LVEF) for the heart, evaluated using ultrasound-based imaging, and fat content using micro-CT, are reported as percentages of controls in C26-bearing mice in blue, in PM01183-treated C26 mice in green and in ET743-treated mice in red (10/group).

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
