# Peer review of "Trabectedin and Lurbinectedin Extend Survival of Mice Bearing C26 Colon Adenocarcinoma, without Affecting Tumor Growth or Cachexia"

_cancers, 2020, doi:10.3390/cancers12082312_

Round 1
Reviewer 1 Report
In the present version of the study the authors have addressed some of the points raised in the previous revision round. However, I'm not sure they really solved the specific concerns. in particular:
- the authors support their results on myogenin expression (increased in mice bearing the C26 tumor) quoting a recent study (ref. 40 in the paper). However, myogenin levels are frequently reduced or unchanged in the muscle of tumor bearing animals, as shown by previous reports (PMID: 21048967; PMID: 24084740; PMID: 30591621). Reduced myogenin content is consistent with the delayed regeneration occurring in cancer cachexia. Indeed, treatments able to improve muscle wasting have also shown to restore the physiological kinetic of Pax7/myogenin expression (PMID: 21048967; PMID: 24084740; PMID: 30209066). By contrast, the authors report increased myogenin levels in mice bearing the C26 tumor, that are counteracted by PM01183 administration, thus raising the following questions: 1) how do the authors explain myogenin increase in the C26 mice?; 2) why myogenin reduction by PM01183 should be beneficial?; 3) in their point-by-point reply the authors state: '...only the induction of myogenin, but not that of C/EBPb, was almost fully abrogated in the PM-treated group. Hence, our data suggest that the treatment counteracts the dysregulated myogenic program in the muscles from C26-bearing mice rather than interfering with the differentiation process'. What does this sentence mean? What do the authors have in mind discriminating between 'counteracts the dysregulated myogenic program' and 'interfering with the differentiation process'?
- the criticisms above are further stressed by the results proposed by the authors in their reply (figure 2). Indeed, they show that the exposure of differentiating myoblasts (days 0-2) to PM01183 leads to reduced Pax7 in face of increased levels of the proliferation marker Ki67. The authors conclude that: 'PM treatment supports myoblast proliferation rather than differentiation'. In this regard, it is accepted that regeneration in cancer cachexia does not depend on impaired satellite cell proliferation, rather, these latter accumulate in the muscle due to delayed differentiation (PMID: 21048967; PMID: 24084740; PMID: 30591621). Along this line, the authors should clarify why a drug able to increase myoblast (satellite cell) proliferation could positively affect muscle regeneration in the C26 mice. As for the lack of Ki67 positivity in muscle sections of tumor bearing mice, it is not surprising to me. Indeed, the muscles examined likely derive from animals euthanized on day 13 or more after tumor transplantation, when the proliferative wave is no more likely to occur;
- the new data on lung metastasis are interesting and should be added to the study, at least as supplemental, since they rule out an important potential contributor to the prolonged survival of PM01183-treated animals. However, the question about the mechanisms underlying such increased survival still remains open. The authors suggest that this could be linked to the anti-splenomegaly and anti-inflammatory action of the drug. It might well be, however it is not demonstrated at all. Rather, only the reduced levels of phosphorylated NF-kB could suggest an effect on inflammation, since the cytokines tested are not affected by the treatment and STAT3-dipendent pathway is equally activated, irrespective of PM01183 administration. As for splenomegaly, the authors correctly talk about myeloid-derived suppressor cells that have been proposed to contribute to cancer cachexia. However, data in Figure S4 show that 'severe extramedullary hematopoiesis characterized by the presence of myeloid cells at different degree of differentiation' can be observed in both treated and untreated C26 mice, which renders the author's hypothesis rather unlikely. I would encourage the authors to make an effort in trying to explain the findings on survival and splenomegaly, at the discussion level, if not any.
Round 2
Reviewer 1 Report
I do not have any additional comment